# Fast Binarized Neural Network Training with Partial Pre-training

## Abstract

Binarized neural networks, networks with weights and activations constrained to lie in a 2-element set, allow for more time- and resource-efficient inference than standard floating-point networks. However, binarized neural networks typically take more training to plateau in accuracy than their floating-point counterparts, in terms of both iteration count and wall clock time. We demonstrate a technique, partial pre-training, that allows for faster from-scratch training of binarized neural networks by first training the network as a standard floating-point network for a short amount of time, then converting the network to a binarized neural network and continuing to train from there. Without tuning any hyperparameters across four networks on three different datasets, partial pre-training is able to train binarized neural networks between $1.26\times$ and $1.61\times$ faster than when training a binarized network from scratch using standard low-precision training.

## 1 Introduction

*Quantizing* neural networks (Gupta et al., 2015), constraining weights and activations to take on values within some small fixed set, is a popular set of techniques for reducing the storage (Han et al., 2016) or compute (Fromm et al., 2020) requirements of deep neural networks. Weights and activations can often be quantized down to as few as 8 bits with no loss in accuracy compared to a full-precision model. Further quantization often comes at the expense of accuracy: it is possible to *binarize* neural networks (Hubara et al., 2016; Rastegari et al., 2016), constraining weights and activations to take on values within a set of two elements (often $\{-1, 1\}$), but such binarization often lowers the accuracy of the resultant network, necessitating a tradeoff between desired compression and accuracy.

In the literature, there are two primary techniques for obtaining a quantized neural network: quantizing a pre-trained full-precision network (Banner et al., 2019; Han et al., 2016), and training a quantized network from scratch (Hubara et al., 2016; Gupta et al., 2015).

**Full-precision training.** Quantizing a full-precision network requires few or even no additional training epochs on top of training that full-precision network. Typical procedures for quantizing a full-precision network range from data-blind procedures like selecting quantization bins to minimize distance from the original weights (Banner et al., 2019), to data-intensive procedures such as retraining the network to be more amenable to quantization (Han et al., 2016). However without significant additional training time, quantizing a pre-trained network often does not reach the highest accuracy possible for the quantized network architecture (Alizadeh et al., 2019). Further, achieving high accuracy with heavy quantization, such as binarization, often requires changing the network architecture, for instance by adding skip connections (Bethge et al., 2019); such architectural changes mean that the weights of a pre-trained full-precision network may not transfer to the new architecture.

**Low-precision training.** Alternatively, training a quantized network from scratch allows for achieving high accuracy regardless of the availability of pre-trained full-precision weights (Alizadeh et al., 2019). Typical procedures for training a quantized network from scratch involve tracking and optimizing *latent weights*, weights which are quantized during the forward pass but treated as full-precision during the backward pass (Hubara et al., 2016). However, training a quantized network from scratch can be costly. Quantized networks typically require more training iterations to plateau in accuracy (Hubara et al., 2016, Figure 1; Bethge et al., 2019, Figure 2). Further, since quantized networks are often trained by simulating the quantized operations in floating-point (Zhang et al., 2019), low-precision training can be even more computationally expensive than the full-precision equivalent.

**Research question.**    In this paper, we explore the question:

*Can we accelerate training a binarized neural network from scratch to a given target accuracy?*

Concretely, we assume that a network architecture and standard training schedule are provided, but that pre-trained full-precision networks are not available. We also specifically focus on achieving accuracy in the early phase of training, exposing the tradeoff between training cost and accuracy.

**Partial pre-training.**    To answer the above research question, we evaluate a technique, *partial pre-training*, that allows for faster training of binarized neural networks by first training the network as a standard floating point network with standard full-precision training for a short amount of time, then converting the network to a binarized neural network and continuing to train from there with standard low-precision training for the remainder of the budgeted training time. We specifically evaluate partial pre-training's speedup over standard low-precision training, when training a binarized neural network from scratch. We find that partial pre-training can train VGG, ResNet, and Neural Collaborative Filtering networks on CIFAR-10, ImageNet, and MovieLens-20m between $1.26\times$ and $1.61\times$ faster than standard low-precision training.

**Contributions.**

- We present partial pre-training, which can train binarized neural networks from scratch between $1.26\times$ and $1.61\times$ faster than standard low-precision training.
- We find that partial pre-training both requires fewer iterations to train to a given accuracy, and also that partial pre-training takes on average less time per iteration than standard low-precision training.
- We analyze the sensitivity of partial pre-training to the choice of split between full-precision and low-precision training finding that an even split, though not always optimal, nearly matches the highest accuracy achievable by any other choice of split.

All together, we find that partial pre-training is a simple and effective approach for accelerating binarized neural network training. Partial pre-training is a step towards the goal of binarized neural network training procedures that can match the efficiency gains of binarized neural network inference.

## 2   BACKGROUND

Binarized neural networks trade off accuracy for inference efficiency. However, binarized neural networks often take longer to train than the full-precision versions of the same network architecture, both in terms of training iterations until convergence and wall-clock time per iteration.

**Training iterations.**    Binarized neural networks tend to take more iterations to train than the full-precision versions of the same network architecture. For instance, Hubara et al. (2016, Figure 1) show a binarized neural network with a custom architecture requiring $4\times$ as many training iterations to plateau in accuracy as a full-precision baseline on CIFAR-10. Bethge et al. (2019, Figure 2) similarly show a binarized ResNet-18 taking $2\times$ as many training iterations to plateau in accuracy as a full-precision baseline on ImageNet.

**Wall-clock time.**    Beyond requiring more iterations to train, binarized neural networks tend to take more wall-clock time to complete each iteration of training than full-precision networks do. This is because binarized neural networks are often trained by simulating low-precision operations with standard floating point operations (Zhang et al., 2019; Fromm et al., 2020), and require additional bookkeeping that full-precision networks do not require, such as performing the binarization of weights and activations (Hubara et al., 2016) and calculating scaling factors (Rastegari et al., 2016). While it is theoretically possible to accelerate binarized neural network training, we are not aware of any effort to exploit binarization during the training phase. It is also not clear what the maximum speedup possible from accelerating binarized neural network training would be: Fromm et al. (2020) show an acceleration of $6.33\times$ for a VGG during inference; with the additional bookkeeping of binarized neural network training and potentially requiring higher precision gradients in the backward pass (Zhou et al., 2016), real training speedups would likely be lower.

## 3 PARTIAL PRE-TRAINING

This paper focuses on accelerating the training time of binarized neural networks. Specifically, we aim to reduce both the training iterations and wall clock time of training binarized neural networks. We achieve this by leveraging the faster training time—both iteration-count and time-per-iteration—of full-precision networks. This section formalizes the design space of partial pre-training algorithms, describing the exact training methodology used in the experiments in Sections 5 and 6.

Partial pre-training splits training into multiple phases. First, partial pre-training trains the network for a short amount of time at full precision (32 bits) using no quantization methods: weights, activations, and gradients are not quantized or clipped. Next, the binarization operators are added to the network (binarizing weights and activations). Partial pre-training then continues to train the binarized neural network using standard low-precision training.

To avoid requiring hyperparameter search for each different network, we prescribe a standard training schedule based on the original training schedule of the full-precision network (i.e., learning rate and associated decay, which is assumed to be provided). Each step of partial pre-training takes $50\%$ of the allotted training time. Within the training time for each step of partial pre-training, the network is trained using the original learning rate schedule compressed to the allotted training time.

The partial pre-training algorithm is presented in Algorithm 1:

---
**Algorithm 1** Partial pre-training.

---
1. Train the network at full precision, using the original learning rate schedule compressed down to half of the desired training time.
2. Binarize the network, inserting quantization operators into the network.
3. Train the binarized network, using the original learning rate schedule compressed down to the remaining half of the desired training time.

---

## 4 EXPERIMENTAL METHODOLOGY

### 4.1 DATASETS AND NETWORKS

We evaluate partial pre-training across a variety of datasets and networks. Specifically, we evaluate partial pre-training on a CIFAR-10 (Krizhevsky, 2009) ResNet-20 (He et al., 2016), a CIFAR-10 VGG-16 (Simonyan & Zisserman, 2014), an ImageNet (Russakovsky et al., 2015) ResNet-34, and a MovieLens 20M (Harper & Konstan, 2015) Neural Collaborative Filtering (NCF) (He et al., 2017) model. Following Bethge et al. (2019), our ResNet-20 and ResNet-34 are extended with additional skip connections past every quantized convolution, rather than past every block of quantized convolutions as is standard for ResNets; this architectural change facilitates training binarized ResNets from scratch. Our NCF model additionally has BatchNorm (Ioffe & Szegedy, 2015) layers after each binarized layer. The networks otherwise use standard architectures, data augmentation schemes, and training schedules drawn from the literature. Details about the networks and their respective training regimes are presented in Table 1.

### 4.2 TRAINING DETAILS

All networks were trained on AWS GPU instances. The CIFAR-10 and MovieLens-20M networks were trained on p3.2xlarge instances, using an NVIDIA V100 GPU. The ImageNet networks were trained on p3.8xlarge instances, training with 4 data-parallel NVIDIA V100 GPUs. The vision networks were trained using custom implementations of each network in TensorFlow 2.2.0 (Abadi et al., 2016). The NCF was trained using the PyTorch 0.4 (Paszke et al., 2019) implementation included in the MLPerf 0.5 training benchmark suite (Mattson et al., 2020).

### 4.3 BINARIZATION

We evaluate partial pre-training using standard approaches to neural network binarization:

| Dataset | Network | Optimizer | Learning rate (t = training iteration) | Test accuracy |
|---|---|---|---|---|
| CIFAR-10 | ResNet-20 | Nesterov SGD $\beta = 0.9$ Batch size: 128 Weight decay: 0.0001 Iterations: 64000 | $\alpha = \begin{cases} 0.1 & t \in [0, 32000) \\ 0.01 & t \in [32000, 48000) \\ 0.001 & t \in [48000, 64000] \end{cases}$ | 32-bit: 90.75% $\pm$ 0.26% 1-bit: 83.30% $\pm$ 0.25% |
| | VGG-16 | | | 32-bit: 93.30% $\pm$ 0.19% 1-bit: 88.52% $\pm$ 0.19% |
| ImageNet | ResNet-34 | Nesterov SGD $\beta = 0.9$ Batch size: 1024 Weight decay: 0.0001 Iterations: 450000 | $\alpha = \begin{cases} 0.4 \cdot \frac{1}{5} & t \in [0, 5) \\ 0.4 & t \in [5, 30) \\ 0.04 & t \in [30, 60) \\ 0.004 & t \in [60, 80) \\ 0.0004 & t \in [80, 90] \end{cases}$ | 32-bit: 71.1% $\pm$ 0.06% top-1 1-bit: 60.9% $\pm$ 0.06% top-1 |
| MovieLens 20M | NCF | Adam $\beta_1 = 0.9$ $\beta_2 = 0.999$ Batch size: 2048 Iterations: 387925[1] | $\alpha = 0.0005$ | 32-bit: 0.593 $\pm$ 0.002 HR@10 1-bit: 0.534 $\pm$ 0.003 HR@10 |

Table 1: Networks, datasets, and hyperparameters. All hyperparameter choices are standard for their respective networks (He et al., 2016; Goyal et al., 2017; Mattson et al., 2020).

**Inputs.** We binarize input activations using PACT (Choi et al., 2018), a gradient-based method of determining activation scale which is used in place of the ReLU activation in binarized neural networks. PACT introduces one trainable parameter per layer, $\alpha$, which controls the scale of the activations. The PACT activation on binarized networks has the following form:

$$\text{PACT}(x) = \begin{cases} 0, & x \in (-\infty, \frac{\alpha}{2}] \\ \alpha, & x \in (\frac{\alpha}{2}, \infty) \end{cases}$$

$$\frac{\partial \text{PACT}(x)}{\partial \alpha} = \begin{cases} 0, & x \in (-\infty, \alpha) \\ 1, & x \in [\alpha, \infty) \end{cases} \qquad \frac{\partial \text{PACT}(x)}{\partial x} = \begin{cases} 1, & x \in [0, \alpha] \\ 0, & \text{otherwise} \end{cases}$$

We initialize $\alpha = 3$ for each layer, and control its magnitude with an L2 regularization penalty with coefficient 0.0002.

**Weights.** We binarize weights using the sign of the weight and the straight-through estimator:

$$\text{sign}(x) = \begin{cases} -1, & x \leq 0 \\ 1, & x > 0 \end{cases} \qquad \frac{\partial \text{sign}(x)}{x} = \begin{cases} 1, & x \in [-1, 1] \\ 0, & \text{otherwise} \end{cases}$$

**Full precision layers.** Following standard practice (Hubara et al., 2016; Simons & Lee, 2019; Qin et al., 2020), we do not binarize inputs or weights to the first, last, or batch normalization layers, nor do we binarize projection layers in ResNets (Bethge et al., 2019).

## 4.4 Experiments and results

For each baseline and configuration of partial pre-training, we run three independent trials, plotting the minimum, median, and maximum accuracy achieved across all three trials.

## 4.5 Speedups

The reported speedups of partial pre-training over low-precision training are calculated by sampling 100 evenly-spaced accuracies, between 50% of the maximum accuracy achievable by the binarized network (as a lower bound on acceptable accuracy) to the accuracy at which the techniques converge (with the exception of the NCF, where low-precision training does not converge to the same accuracy as partial pre-training). Speedups are then calculated by determining the time to train each technique to that accuracy (linearly interpolating between nearby points if there is no trial at exactly that accuracy), calculating the speedup as $\frac{\text{low-precision training time}}{\text{partial pre-training training time}}$, and calculating the harmonic mean of the speedup across all target accuracies.

---

[1]The source implementation (from MLPerf) trains the NCF to an accuracy threshold of 0.635 HR@10, taking a mean duration of 388000 iterations (Mattson et al., 2020, Figure 3a). Adding BatchNorm slightly decreases the accuracy of the full-precision network, but is necessary to train the binarized network.

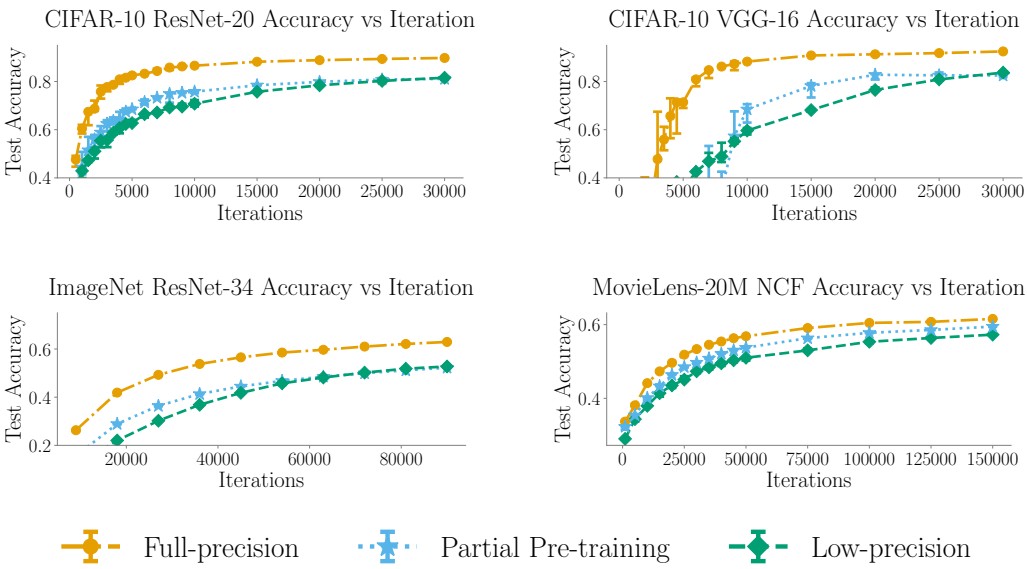

Figure 1: Accuracy versus training iterations of partial pre-training and baselines.

## 5 EXPERIMENTAL RESULTS

This section presents the accuracy achieved by partial pre-training across training times on several different networks. Section 5.1 shows the accuracy achieved by partial pre-training when training for different numbers of training iterations, finding that iteration-for-iteration partial pre-training trains faster than standard low-precision neural network training. Section 5.2 extends the analysis to compare wall-clock time, finding that the overhead of simulated low-precision training leads to partial pre-training training even faster than low-precision training.

### 5.1 ACCURACY V.S. TRAINING ITERATION

We find that partial pre-training can accelerate binarized neural network training iteration-for-iteration: binarized neural networks trained with partial pre-training require fewer training iterations to reach any target accuracy than binarized neural networks trained with standard low-precision training. Among the four networks presented in this section, we find a mean speedup of $1.30\times$ of partial pre-training over standard low-precision training when compared iteration-for-iteration.

**Methodology.** Each plot shows the accuracy attainable when training a neural network for the specified number of training iterations $t$. Full-precision is the accuracy attained from training a full-precision network from scratch for the specified number of iterations, and is presented as an upper bound on training speed. Partial pre-training is the accuracy of the partial pre-training procedure. Low-precision is the accuracy attained from training a binarized network from scratch using standard low-precision training for the specified number of iterations. As specified in Section 4.4, each point on the graph is independent, and shows the median and min/max accuracy from three trials.

**Results.** Figure 1 presents the iteration-for-iteration accuracy of partial pre-training compared to full-precision and low-precision training. We find that partial pre-training matches or exceeds the accuracy of low-precision training for any training time, across all networks and datasets, with the exception of after fewer than 10000 training iterations of the CIFAR-10 VGG-16. This speedup is most prominent with relatively little training time, but tapers off and eventually disappears when the network is trained for enough iterations with standard low-precision training; more analysis of this behavior is provided in Section 6.

| Dataset | Network | Full-precision training speed | Low-precision training speed | Low-precision slowdown |
|---------|---------|------------------------------|------------------------------|------------------------|
| **CIFAR-10** | **ResNet-20** | 79.86 it/s | 64.04 it/s | 1.25× |
| **CIFAR-10** | **VGG-16** | 57.16 it/s | 43.37 it/s | 1.32× |
| **ImageNet** | **ResNet-34** | 9.12 it/s | 7.08 it/s | 1.29× |
| **MovieLens-20M** | **NCF** | 103.75 it/s | 72.88 it/s | 1.42× |

Table 2: Slowdown of low-precision compared to full-precision training.

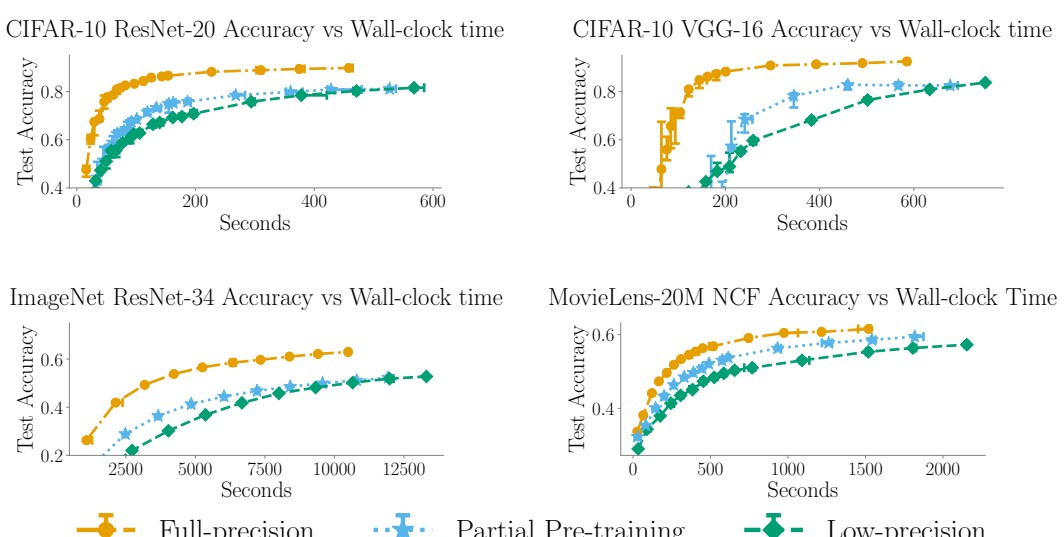

Figure 2: Accuracy versus wall-clock time of partial pre-training and baselines. Same as Figure 1, but with wall-clock time on the x axis rather than training iterations

## 5.2 ACCURACY V.S. WALL-CLOCK TIME

This section presents the accuracy attained by partial pre-training across different wall-clock training times. Table 2 first presents the per-training-iteration slowdown of simulated low-precision optimization, the standard practice in the literature. Figure 2 then presents the accuracy attained at different wall-clock times, showing how the per-iteration accuracy gains from Section 5.1 and the full-precision speedups from Table 2 compound to lead to faster network training from partial pre-training.

**Methodology.** The training speeds in Table 2 are calculated by timing the mean time to complete a training epoch across 5 epochs, ignoring the first 3 batches in each epoch. The speeds are presented in training iterations per second, with each iteration corresponding to processing one batch of data according to the batch sizes presented in Table 1.
The data in Figure 2 are from the same training runs as Figure 1, but with wall-clock time rather than iteration count presented on the x axis. Wall clock time is measured from just before the start of the first iteration of training until just after the last iteration of training, and is inclusive of any overhead from the full-precision to low-precision projection in partial pre-training.

**Results.** Figure 2 presents the end-to-end wall-clock time of partial pre-training and baselines. Due to the combination of more accuracy per iteration from partial pre-training (Figure 1) and the faster training time of the full-precision training phase (Table 2), we find that partial pre-training trains binarized networks faster than standard low-precision training inclusive of all overhead from the partial pre-training process. Specifically, in the range of data shown in Figure 2, we find that partial pre-training has a harmonic mean speedup over standard low-precision training of: 1.28× for the CIFAR-10 ResNet-20, 1.26× for the CIFAR-10 VGG-16, 1.36× for the ImageNet ResNet-34, and 1.61× for the MovieLens-20M NCF.

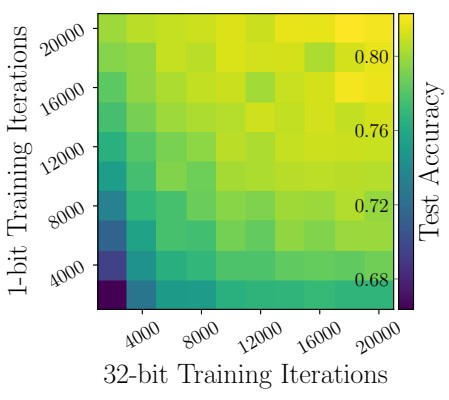 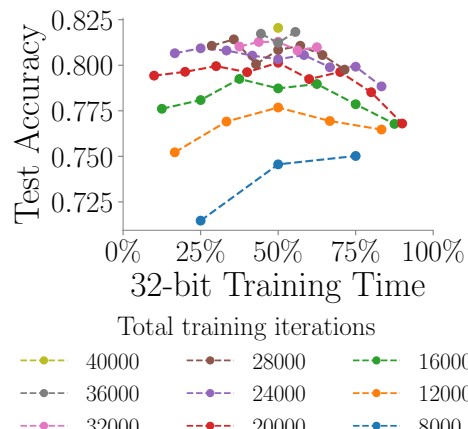

(a) Accuracy when training with varying amounts of full-precision and low-precision iterations.

(b) Accuracy when training with equivalent training time but varying full-precision/low-precision split.

Figure 3: Sensitivity of partial pre-training to different combinations of full-precision training and low-precision training times.

## 6    ANALYSIS

Section 5 demonstrated that partial pre-training allows for faster training of binarized neural networks than standard low-precision training. This section analyzes the hyperparameter choice baked into the partial pre-training algorithm of splitting the training time evenly between full-precision and low-precision training, finding that an even split between full-precision and low-precision training allows for generally good performance across training times. Due to compute limitations, this section only analyzes the CIFAR-10 ResNet-20. We find that while this even split is not optimal in all circumstances, it is a generally applicable hyperparameter choice that leads to accuracy competitive with the best choice of training time split within the given training budget.

**Methodology.**    Figure 3a shows the accuracy from training a CIFAR-10 ResNet-20 with a modification of partial pre-training that has a non-even split between full-precision and low-precision training. Each point is the median accuracy across 3 trials when first training for the number of iterations specified on the x-axis with the original learning rate schedule compressed to the number of training iterations, then projecting to low-precision and training for the number of iterations specified on the y-axis again with the original learning rate schedule compressed to the number of training iterations. Figure 3b is generated from the same data as Figure 3a, showing the accuracy along the top-left to bottom-right diagonals of Figure 3a. That is, Figure 3b shows partial pre-training's accuracy as the split between full-precision and low-precision training is varied for a given training budget, with each different line representing a different training budget.

**Results.**    Figure 3 shows that the even split between full-precision and low-precision training baked into partial pre-training is a generally applicable hyperparameter choice that leads to accuracy competitive with the best choice of training time split within the given training budget. When training for a short duration (leading to lower overall accuracy, shown in the lower lines in Figure 3b), spending more time on the full-precision training phase leads to higher accuracy than spending more time on the low-precision training phase. When training for a longer duration (shown in the higher lines), spending more time on the low-precision phase leads to higher accuracy than spending more time on the full-precision phase. Across all training budgets, splitting time evenly between full-precision and low-precision training leads to competitive accuracy with the best choice of hyperparameters.

## 7    LIMITATIONS

While we find that partial pre-training accelerates binarized neural network training time across the networks and datasets experimented on in Section 5, there are dimensions of the hyperparameter space that we do not explore. We do not evaluate binarization techniques other than the commonly

used PACT activation scaling (Choi et al., 2018) and static weight scaling. We also do not evaluate learning rate schedules other than compressing the original learning rate schedule from the fist phase of training; it is plausible that other learning rate schedules better tuned to the technique or network could train even faster (Renda et al., 2020), but we do not evaluate these alternative schedules.

## 8 RELATED WORK

Several other papers discuss related ideas to partial pre-training. However, this paper is the first to propose partial pre-training as a method for training binarized neural networks from scratch, to provide a concrete partial pre-training algorithm, and to systematically evaluate partial pre-training as a method of speeding up the training of binarized neural networks.

**Faster binarized network training.** Partial pre-training's speedup is due to the slow training speed of low-precision network training, both in terms of iteration count and wall-clock speed per iteration; partial pre-training would be an ineffective training method given a method that could train low-precision networks at an equivalent or faster rate than the full-precision counterparts. Here we survey techniques that propose speeding up low-precision network training, showing that partial pre-training is not supplanted by better low-precision training techniques.
Zhou et al. (2016) propose a technique that uses low-bit quantization in the backward pass in addition to the forward pass, but neither measure the iteration-count slowdown induced by gradient quantization nor measure well-clock speedup of gradient quantization. De Sa et al. (2018) propose a technique to reduce quantization error between latent and quantized weights to improve asymptotic training speed; however, the proposed algorithm is only evaluated on 8-bit networks. Alizadeh et al. (2019) suggest that commonly applied weight and gradient clipping techniques constrain the learning rate that can be used during training, and propose disabling gradient clipping to allow for larger learning rates and therefore faster training; however, they do not disable the quantization during training (still incurring overhead from quantization operators in the training graph), they do not evaluate the wall-clock speedup of the approach, and further the approach requires bespoke tuning of the accelerated training schedule per-network.

**Full-precision pre-training.** Alizadeh et al. (2019) and Bethge et al. (2019) both find that binarizing a fully trained full-precision network to low-precision then training the binarized network for a short amount of time allows for rapid re-training of the low-precision network. However, Alizadeh et al. and Bethge et al. dismiss the approach because the low-precision network that is fine-tuned for a short amount of time does not achieve the same accuracy as a low-precision network trained from scratch for a longer duration. Alizadeh et al. and Bethge et al. also only evaluate fine-tuning from fully trained full-precision networks, rather than the partially-trained full-precision networks presented in this paper, and do not evaluate the wall-clock speedups of such an approach.

**Multi-phase binarized network training.** Zhou et al. (2017) propose an approach that incrementally quantizes neural networks by partitioning weights into disjoint groups based on the magnitude of the pre-trained weights, first binarizing weights with large magnitudes then binarizing weights with small magnitudes. Zhou et al. start this process from a fully-trained full-precision network, and do not consider observed training speedups, only accuracy gains. Verhoef et al. (2019) gradually reduce the number of bits used for weight representation over the course of training, with the goal of improving accuracy; Verhoef et al. do not evaluate the training speedup achieved by the approach, nor does their technique allow for taking advantage of faster full-precision training.

## 9 CONCLUSION

Binarizing neural networks makes them more efficient for inference, but binarized neural networks trained from scratch typically take more iterations and more wall-clock time to plateau in accuracy than their full-precision counterparts. In this paper we demonstrate a technique, partial pre-training, that allows for faster training of binarized neural networks by exploiting fast full-precision training then fine-tuning at low-precision. Without introducing any new hyperparameters, we find that partial pre-training can train VGG, ResNet, and NCF networks on CIFAR-10, ImageNet, and MovieLens-20m between $1.26\times$ and $1.61\times$ faster than standard 1-bit training. All together, we find that partial pre-training is a simple and effective approach for accelerating binarized neural network training. Partial pre-training is a step towards the goal of binarized neural network training procedures that can match the efficiency gains of binarized neural network inference.

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
