# OpenReview forum: "Fast Binarized Neural Network Training with Partial Pre-training"
_ICLR.cc/2021/Conference — Reject_

### Official Review · AnonReviewer2 · 2020-10-22
**Little novelty and weak results**

**Rating:** 4
**Confidence:** 5

**Review:**

The paper suggests a method for training binary neural networks. The proposed method is to partially train with full precision and then continue with binarized training using the straight-through estimator. The method is very simple and there is very limited technical contribution, so in order to be worthy of publication it needs to be supported with compelling experimental results. Unfortunately this is not the case.

The main claim is speeding up training by a factor of 1.2/1.6. While this can help the importance of speeding up training (unless by a much larger factor) is quiet limited. The useful speedup BNN present is at inference time, and the time to train a network is of much less importance (at least for this kind of speedup).
From Fig.1 I am not convinced that the speedup claims hold. You can see the test accuracy for binary training on 3 of the experiments reaches the Partial Pre-training level but it doesn't look like it completely flattened out yet. It looks like if you want take the best model on test, then you don't get any speedup
Results on cifar-10 seem quiet poor (both full precision and 1-bit). For example "Learning discrete weights using the local reparametrization trick" gets 93.2 1-bit acc on cifar-10 with VGG.

---

> ### Author Response · Authors · 2020-11-17
> **Response to Reviewer 2**
>
> Thank you for feedback and commentary! We have provided a general response to all reviewers above ([https://openreview.net/forum?id=H6ZWlQrPGS2&noteId=h4-wXzhNMG](https://openreview.net/forum?id=H6ZWlQrPGS2&noteId=h4-wXzhNMG)). Here we will provide response to your individual points.
>
> > The main claim is speeding up training by a factor of 1.2/1.6. While this can help the importance of speeding up training (unless by a much larger factor) is quiet limited. The useful speedup BNN present is at inference time, and the time to train a network is of much less importance (at least for this kind of speedup).
>
> Please see the discussion of significance of training speedup in the general response above.
>
> > It looks like if you want take the best model on test, then you don't get any speedup...
>
> Please see the discussion of final accuracy in the general response above.
>
> > Results on cifar-10 seem quiet poor (both full precision and 1-bit)...
>
> The accuracies reported for CIFAR-10 are standard for the network and hyperparameters used in the evaluation [7; Tables 3,4]. Partial pre-training can be applied with an arbitrary low-precision training setup (including other quantization approaches or other training schemes like that of [8]) in the second phase, though validation of this for other quantization approaches is left to future work. Please also see the discussion of hyperparameter choices and sweeps in the general response above.
>
> ### Citations
> [7] "Binary Neural Networks: A Survey," Haotong Qin, Ruihao Gong, Xianglong Liu, Xiao Bai, Jingkuan Song, Nicu Sebe.

---

### Official Review · AnonReviewer4 · 2020-10-26
**Providing a heuristic method to speed up low-precision training**

**Rating:** 4
**Confidence:** 5

**Review:**

This paper proposed one simple method called partial pre-training to speed up the training of binary neural networks (BNN). The pros and cons are as follows:

Pros:
1. The partial pre-training method is simple and easy to implement;
2. For standard binary optimizer like the straight-through-estimator (STE), the method improves the training speed to some extent;

Cons:
1. The main concern of the proposed method is kind of heuristic and there is a lack of theoretical explanation, whether rigorous or not, why this method works.

2. As described in Section 7 by the author themselves, there are several apparent limitations of current evaluation, e.g., several dimensions of the hyper-parameters are not explored, other Binary optimizers are not considered, different learning rate schedules, etc. As a result, it is unconvinced that the partial pre-training could universally improve the speed as a general method. In addition, the improvement of speed-up are not very apparent especially for ResNet-20 and ResNet-34 as shown in Fig. 1 and Fig. 2, i.e., it took approximately the same time to reach the final precision even though the proposed method achieves higher accuracy before saturation. Given the inadequate evaluations and lack of theoretical explanations, this might be due to unfair comparison.

3. There is a lack of explanation of the contradiction result with previous result proposed in Alizadeh et al (2019), which dismissed the approach of pre-training. Is there good explanation of such an opposite result? It would be better to show results of different split between full-precision training and low-precision training.

4. Regarding pre-training for BNN, there are some related works from the Bayesian perspective. In Shayer et al. (2018), they used the result of full-precision training as the prior for the binary training, which improves the final result, as opposed to Alizadeh et al (2019). The Bayesian perspective provides an explanation of the effectiveness of a good prior. In Meng et al. (2020), they showed that STE could be viewed as Bayesian and obtained good result even with a uniform prior. Also, the posterior obtained after full-training (binary) could be used as prior to enable continual learning, which shows effectiveness of the prior. Given the above results, since the authors demonstrate that partial pre-training can increase the speed for STE, does this imply that partial pre-training provides a better prior than full pre-training? If so, why?

Shayer, O., Levi, D., and Fetaya, E. Learning discrete weights using the local reparameterization trick. ICLR, 2018.
Meng, X, Bachmann. R., Khan. E. Training Binary Neural Networks using the Bayesian Learning Rule. ICML, 2020.

---

> ### Author Response · Authors · 2020-11-17
> **Response to Reviewer 4**
>
> Thank you for feedback and commentary! We have provided a general response to all reviewers above ([https://openreview.net/forum?id=H6ZWlQrPGS2&noteId=h4-wXzhNMG](https://openreview.net/forum?id=H6ZWlQrPGS2&noteId=h4-wXzhNMG)). Here we will provide response to your individual points.
>
> > The main concern of the proposed method is kind of heuristic and there is a lack of theoretical explanation, whether rigorous or not, why this method works.
>
> Please see the discussion of theoretical analysis in the general response above.
>
> > There is a lack of explanation of the contradiction result with previous result proposed in Alizadeh et al (2019), which dismissed the approach of pre-training. Is there good explanation of such an opposite result?
>
> The results compare the accuracies of each technique when training the network from random initialization with a fixed training budget. Compared to standard low-precision training, partial pre-training is therefore allotted half as much time in each phase, meaning the methods are compared at equal training budgets. We will clarify this in the final version of the paper.
>
> > It would be better to show results of different split between full-precision training and low-precision training.
>
> Results for different splits between full-precision training and low-precision training are presented and discussed in Section 6.
>
> > Regarding pre-training for BNN, there are some related works from the Bayesian perspective... [D]oes this imply that partial pre-training provides a better prior than full pre-training?
>
> The connection to the Bayesian perspective of [8,9] is an interesting connection to explore in future work. To clarify the exact relationship between our claims and the analysis of [9], our results do not imply that partial pre-training provides a better prior than full pre-training. Figure 3a shows that for a fixed amount of low-precision training (i.e., elements on any given row) a larger amount of full-precision training (i.e., farther to the right) results in higher accuracy.
>
> ### Citations
>
> [8] "Learning Discrete Weights Using the Local Reparameterization Trick," Oran Shayer, Dan Levi, Ethan Fetaya.
>
> [9] "Training Binary Neural Networks using the Bayesian Learning Rule," Xiangming Meng, Roman Bachmann, Mohammad Emtiyaz Khan.

---

> > ### Comment · AnonReviewer4 · 2020-11-24
> > **Thanks for the response!**
> >
> > Thanks a lot for the authors' response, which addresses part of my previous concerns.
> >
> > For the theoretical problem, I agree that experiments alone are important and rigorous analysis could come afterwards. As shared by R1, current experiments seem not enough for claiming "a rigorous empirical evaluation" due to the lack of universality evaluation.  I am kind of concerned about this point.

---

> > > ### Author Response · Authors · 2020-11-25
> > > **Re: empirical evaluation**
> > >
> > > The concern about applicability to other binary optimizers is a reasonable point, and indeed one that our experiments do not answer. To avoid generating potentially misleading conclusions about the performance of the method on binary optimizers not fully evaluated in the work, we will add clear statements to the introduction that our results are only validated on optimizing latent weights with SGD and ADAM.
> > >
> > >
> > > Regarding other hyperparameters ("different learning rate schedules, etc"), our evaluation includes 4 different networks on 3 different datasets. Each dataset uses a different learning rate schedule, batch size, and other network hyperparameters. While we agree in principle that more validation across hyperparameters would further validate our findings, this further validation is beyond the scope of this paper.

---

### Official Review · AnonReviewer3 · 2020-10-28

**Rating:** 5
**Confidence:** 4

**Review:**

#### Comments
Summary:

The authors propose a fast binarized neural network training algorithm that splits the whole training process into the full precision training stage and the binary training stage. The experimental results show some improvement about training speed in terms of iterations and wall clock time. Generally, the paper is well written.

Strength:
--The idea is reasonable and the method is presented clearly.
--The experimental results indicate that the proposed method can accelerate the convergence of the binary networks on image classification and collaborative filtering.

Weakness:
--The comparison between the proposed method and quantization-finetuning method is lacked, which seems like a closely related work.
--The analysis of the proposed method should be also enhanced. The reason why partial pretraining can improve the training of binary neural networks should be investigated more deeply.

Comments:
(1) The authors claim that the proposed method allows for faster from-scratch training of binarized neural. This seems contradictory to the partial pretraining. The authors may provide more discussion and clarification.
(2) The improvement of the proposed method is not significant. The proposed algorithm speeds up the training marginally and cannot improve the final test accuracy. This limits the contribution.

Overall, the reviewer doesn’t recommend accepting this manuscript at its form. The author may demonstrate more differences between the proposed method and the standard quantization-finetuning method.

---

> ### Author Response · Authors · 2020-11-17
> **Response to Reviewer 3**
>
> Thank you for feedback and commentary! We have provided a general response to all reviewers above ([https://openreview.net/forum?id=H6ZWlQrPGS2&noteId=h4-wXzhNMG](https://openreview.net/forum?id=H6ZWlQrPGS2&noteId=h4-wXzhNMG)). Here we will provide response to your individual points.
>
> > The comparison between the proposed method and quantization-finetuning method is lacked
>
> Based on our understanding of this question (quantizing a pre-trained network then fine-tuning it for a relatively short duration), we do compare against this approach in Section 6, Figure 3. Please see the discussion of other splits between full-precision training and low-precision training in the general response.
>
> > The analysis of the proposed method should be also enhanced.
>
> Please see the discussion of theoretical and causal results in the general response above.
>
> > The authors claim that the proposed method allows for faster from-scratch training of binarized neural. This seems contradictory to the partial pretraining. The authors may provide more discussion and clarification.
>
> The results compare the accuracies of each technique when training the network from random initialization with a fixed training budget. Compared to standard low-precision training, partial pre-training is therefore allotted half as much time in each phase, meaning the methods are compared at equal training budgets. We will clarify this in the final version of the paper.
>
> > The improvement of the proposed method is not significant. The proposed algorithm speeds up the training marginally and cannot improve the final test accuracy. This limits the contribution.
>
> Please see both the discussions of significance of training speedup and also of final accuracy in the general response above.

---

### Official Review · AnonReviewer1 · 2020-10-28
**More comparisons and longer training runs will make the result more impressive**

**Rating:** 4
**Confidence:** 3

**Review:**


This paper addresses the problem of slower training speed with low-precision training of neural nets. It presents a simple solution: first train with full precision on half of the budgeted trainining time, then train with low pecision in the remaining time. This achieves 1.2x - 1.6x speedup compared to low-precision training.

Pro:
- The proposed idea is simple and it is nice to see that it works

Cons:
- I feel there is not enough content (in terms of ideas and experiments) to warrant a full paper. Compared to other ICLR papers, the contributions seem on the low side. See suggestions below.
- The paper mainly compares the proposed method with low-precision training, but the results would be stronger if also compared with full-precision training followed by quantization. This is especially because all the low-precision accuracies trail behind the full precision ones in the results.

Suggestions:
- Experiment with more quantization methods. Currently we do not know whether the proposed method is uniquely suited to the PACT method used, or is a general technique.
- A 1.2x-1.6x speedup on a training process that takes 600 seconds (e.g. CIFG-10 result in Fig 2) seems not so impactful in the grander scheme of things. Even the 12500 second training time is just <4 hours. I understand the results should transfer, but the results would be more impressive if done on larger training runs.

Minor questions/comments:
- Please comment on the terminology. Is what you call low precision training similar to the mixed-precision training now implemented in TensorCore NVIDIA GPUs?
- Table 2: what is it/s. Is it iterations per second?
- Table 1 shows learning rate schedule for t up to 60k in CIFAR or 450k in ImageNet, but Figure 1 stops way before those points in the x-axis. This was somewhat confusing.
- Another clarification point about Fig 1 and 2: for the proposed method, is the full precision training part of the time included in the calculation? I believe so but just want to double-check.

---

> ### Author Response · Authors · 2020-11-17
> **Response to Reviewer 1**
>
> Thank you for feedback and commentary! We have provided a general response to all reviewers above ([https://openreview.net/forum?id=H6ZWlQrPGS2&noteId=h4-wXzhNMG](https://openreview.net/forum?id=H6ZWlQrPGS2&noteId=h4-wXzhNMG)). Here we will provide response to your individual points.
>
> > The paper mainly compares the proposed method with low-precision training, but the results would be stronger if also compared with full-precision training followed by quantization...
>
> Based on our understanding of this question (quantizing a pre-trained network then fine-tuning it for a relatively short duration), we do compare against this approach in Section 6, Figure 3. Please see the discussion of other splits between full-precision training and low-precision training in the general response.
>
> > Experiment with more quantization methods...
>
> Please see the discussion of hyperparameter choices and sweeps in the general response above.
>
> > A 1.2x-1.6x speedup on a training process ... seems not so impactful in the grander scheme of things
>
> Please see the discussion of significance of training speedup in the general response above.
>
> > Is what you call low precision training similar to the mixed-precision training now implemented in TensorCore NVIDIA GPUs?
>
> What we define as low-precision training (training with weights constrained to lie in a small fixed set) is possible to accelerate in some cases in NVIDIA Tensor Cores, though not always. Specifically, according to [https://developer.nvidia.com/blog/tensor-cores-mixed-precision-scientific-computing/](https://developer.nvidia.com/blog/tensor-cores-mixed-precision-scientific-computing/) NVIDIA Tensor Cores can accelerate low-precision training when the weights are represented by as few as 8 bits. In this paper, we consider low-precision training with 1 bit, which NVIDIA Tensor Cores do not have native support for accelerating. As noted in Section 2, we are not aware of any existing results that show faster wall-clock speeds for training binarized neural networks.
>
> > Table 2: what is it/s. Is it iterations per second?
>
> Correct, this is iterations per second. We will clarify this in the final version of the paper.
>
> > Table 1 shows learning rate schedule for t up to 60k in CIFAR or 450k in ImageNet, but Figure 1 stops way before those points in the x-axis. This was somewhat confusing.
>
> Each point on the plot shows a network trained for the specified amount of time using a learning rate schedule which is compressed down to that amount of time, as specified in Algorithm 1. We will clarify this in the final version of the paper.
>
> > ...is the full precision training part of the time included in the calculation?
>
> Correct, full precision training time is included in the calculation. Each point on the plot shows the accuracy when training from random initialization within the given budget.

---

> > ### Comment · AnonReviewer1 · 2020-11-20
> > **thanks**
> >
> > Thanks for the various clarifications. Regarding the author's "general response" to all reviewers, my replies will be placed separately in that thread.

---

### Author Response · Authors · 2020-11-17
**General response to all reviewers (pt 1)**

Thanks to all of the reviewers for their feedback. Here we provide responses to common questions and concerns raised by reviewers. We also provide individualized responses to each reviewer as a comment to each reviewer.

### Significance of training speedup (Reviewers 2, 3)

> Reviewer 2: "The useful speedup BNN present is at inference time, and the time to train a network is of much less importance..."

> Reviewer 3: "The improvement of the proposed method is not significant. The proposed algorithm speeds up the training marginally... This limits the contribution."

The claim that the speedup exhibited by partial pre-training is too small or not important is a subjective claim that does not accord with the interest in such techniques in academia and industry. ICLR has accepted and awarded papers that propose techniques to speed up the training of efficient neural networks, including papers that provide no evaluation of wall-clock time [1, 2, 3]. Though we agree that motivation for acquiring efficient neural networks (e.g., BNNs) is to speed up inference, the cost of training such networks is still a relevant concern when training costs are large [4], especially given that BNNs train less efficiently than full-precision neural networks [3,5]. Our paper identifies a clear problem and makes a tangible step towards solving it, reducing the cost of acquiring binarized neural networks.

### Final accuracy (Reviewers 2, 3, 4)

> Reviewer 2: "It looks like if you want take the best model on test, then you don't get any speedup"

> Reviewer 3: "The improvement of the proposed method is not significant. The proposed algorithm ... cannot improve the final test accuracy..."

> Reviewer 4: "it took approximately the same time to reach the final precision even though the proposed method achieves higher accuracy before saturation"

As discussed in the introduction and methodology sections, we focus our analysis on the region early in training that exposes the tradeoff between accuracy and training cost, similar to the tradeoffs posed by BNNs themselves. Partial pre-training does not result in a higher final test accuracy, and we do not claim it does in the paper; instead, it results in a more Pareto optimal tradeoff curve between training cost and resultaning accuracy.

### Hyperparameter choices and sweeps (Reviewers 1, 4)

> Reviewer 1: "Experiment with more quantization methods. Currently we do not know whether the proposed method is uniquely suited to the PACT method used, or is a general technique."

> Reviewer 4: "several dimensions of the hyper-parameters are not explored, other Binary optimizers are not considered, different learning rate schedules, etc."

We have already validated partial pre-training across multiple datasets, networks, optimizers, learning rate schedules, and other training hyperparameters parameters as noted in Table 1. All of our hyperparameter choices, including choices of quantization methods, are standard in literature and practice and do not require additional hyperparameters that must be searched over or validated across. Given the cost of generating the plots in Figure 1, validation or repudiation of the proposed technique in other settings beyond the standard settings presented in the paper is beyond the scope of this paper, and is therefore left for future work.

### Theoretical and causal results (Reviewers 3, 4)

> Reviewer 3: "The reason why partial pretraining can improve the training of binary neural networks should be investigated more deeply."

> Reviewer 4: "The main concern of the proposed method is kind of heuristic and there is a lack of theoretical explanation, whether rigorous or not, why this method works."

Our contributions include the partial pre-training algorithm along with a rigorous empirical evaluation and analysis of the proposed algorithm. Rigorous theoretical results and causal analysis are left for future work, as is often the case in empirical papers at ICLR [1,6].

---

> ### Author Response · Authors · 2020-11-17
> **General response to all reviewers (pt 2)**
>
> ### Other splits between full-precision training and low-precision training (Reviewers 1, 4)
>
> > Reviewer 1: "The paper mainly compares the proposed method with low-precision training, but the results would be stronger if also compared with full-precision training followed by quantization."
>
> > Reviewer 4: "It would be better to show results of different split between full-precision training and low-precision training."
>
> We do compare against full-precision training followed by quantization in Section 6, Figure 3. Specifically, the right half of Figure 3(b) shows instances of training a network at full-precision for the majority of the budgeted training time, then quantizing and fine-tuning it for a small amount of additional training time. We find that partial pre-training results in equivalent or higher accuracies in general than this technique on a CIFAR-10 ResNet-20. We will include equivalent comparisons on all networks in the final version of the paper.
>
> ### Citations
>
> [1] "The Lottery Ticket Hypothesis," Jonathan Frankle and Michael Carbin.
>
> [2] "SNIP: Single-shot Network Pruning based on Connection Sensitivity," Namhoon Lee, Thalaiyasingam Ajanthan, Philip H. S. Torr.
>
> [3] "An Empirical study of Binary Neural Networks' Optimisation", Milad Alizadeh, Javier Fernández-Marqués, Nicholas D. Lane, Yarin Gal.
>
> [4] "Energy and Policy Considerations for Deep Learning in NLP," Emma Strubell, Ananya Ganesh, Andrew McCallum.
>
> [5] "Binarized Neural Networks: Training Deep Neural Networks with Weights and Activations Constrained to +1 or -1," Matthieu Courbariaux, Itay Hubara, Daniel Soudry, Ran El-Yaniv, Yoshua Bengio.
>
> [6] "Deep Compression: Compressing Deep Neural Networks with Pruning, Trained Quantization and Huffman Coding," Song Han, Huizi Mao, William J. Dally.

---

> > ### Comment · AnonReviewer1 · 2020-11-20
> > **Regarding Other splits between full-precision training and low-precision training**
> >
> > I re-read the explanation for Fig 3b and and not sure if it answers my question. It's possible we're talking about different things, or that maybe I'm misinterpreting the figure. My understanding fo Fig 3b is that it is a different view of Fig 3a, which basically varies the amount of full-precision vs low-precision training in your proposed algorithm.
> >
> > When I said "results would be stronger if also compared with full-precision training followed by quantization," I'm referring not to your proposed algorithm but to another baseline. This baseline trains in full-precision, saves the model, and then post-hoc quantizes it to a binary network. Basically it's just a baseline I might run if I didn't know about your method, and am just interested in the accuracy difference. Maybe it's just a matter of taking your full-precision models in Figure 2 (in orange) and then quantizing it. You might have these numbers but I couldn't figure it out from the paper.
> >
> > (I guess this might correspond to the 100% point in Fig 3(b) but there are no points at that region in the plot.)

---

> > > ### Author Response · Authors · 2020-11-25
> > > **Re: Regarding Other splits**
> > >
> > > You are correct that this corresponds to the 100% point in Fig 3(b), which we did not include in our original set of experiments. The trend in Fig 3(b) is that past a roughly 50/50 split between full-precision and low-precision training, more full-precision training time results in lower accuracy than more low-precision training time; we should expect that this trend will continue to 100% full-precision training time, replicating the results in the literature that low-precision training tends to result in higher accuracy than full-precision training followed by quantization [3,10]. We will include an evaluation of this baseline, specifically an extension of Figure 3 to full-precision training followed by quantization (i.e., the points at 100%) in the final version of the paper.
> > >
> > > ### Citations
> > >
> > > [3] "An Empirical study of Binary Neural Networks' Optimisation", Milad Alizadeh, Javier Fernández-Marqués, Nicholas D. Lane, Yarin Gal.
> > >
> > > [10] "Back to Simplicity: How to Train Accurate BNNs from Scratch?" Joseph Bethge, Haojin Yang, Marvin Bornstein, Christoph Meinel.

---

> ### Comment · AnonReviewer2 · 2020-11-18
> **Regarding previous work on speedup**
>
> I am less familiar with [2,3] but the work in [1] is not mainly about speeding up training time. The only discuss training time costs in the appendix! The work is about learning sparse networks with high accuracy, that was there main claim and what they showed in the experiments. This is completely unrelated to this work where the claim is not that you find a better network, or same performance on a faster network but that you train a fast network faster.

---

> > ### Author Response · Authors · 2020-11-20
> > **Regarding previous work on speedup**
> >
> > > the work in [1] is not mainly about speeding up training time... The work is about learning sparse networks with high accuracy...
> >
> > Sparse networks with high accuracy could have been obtained by training the full network for the full duration of training then pruning after training, as is standard practice [8, 9]. [1] shows that at initialization there exist sparse subnetworks that can train to match the accuracy of the full network, with the explicit stated motivation of improving training performance:
> >
> > > Abstract: "However, if a network can be reduced in size, why do we not train this smaller architecture instead in the interest of making training more efficient as well?"
> >
> > > Introduction: "However, contemporary experience is that the sparse architectures produced by pruning are difficult to train from the start, which would similarly improve training performance."
> >
> > > Introduction: "Improve training performance. Since winning tickets can be trained from the start in isolation, a hope is that we can design training schemes that search for winning tickets and prune as early as possible."
> >
> > That [1] does not measure training time costs is precisely why we cite it as an example of a paper that proposes speeding up training of efficient neural networks that does not evaluate wall-clock time. Of course, [1] is not a direct parallel to our paper, with claims and contributions that do not overlap with those of our paper; [1] is just one of several ICLR papers that include the motivation of lowering training costs for efficient neural networks, but that do not show explicit wall-clock speedups.
> >
> > ### Citations
> > [1] "The Lottery Ticket Hypothesis," Jonathan Frankle and Michael Carbin.
> >
> > [8] "Learning both weights and connections for efficient neural networks," Song Han, Jeff Pool, John Tran, William J. Dally.
> >
> > [9] "Comparing Rewinding and Fine-tuning in Neural Network Pruning," Alex Renda, Jonathan Frankle, Michael Carbin.

---

> > > ### Comment · AnonReviewer2 · 2020-11-22
> > > **A few things to note**
> > >
> > > First, [8] was not cited in [1] and [9] is a later work, so [1] did not treat [8,9] as a baseline to improve upon.
> > >
> > >  Regarding the first quote, it is misleading as you need to look into the next sentence
> > > "However, if a network can be reduced in size, why do we
> > > not train this smaller architecture instead in the interest of making training more efficient as well?
> > > Contemporary experience is that the architectures uncovered by pruning are harder to train from the
> > > start, reaching lower accuracy than the original networks"
> > >
> > > The point isn't about speeding up training, it is that the fast method leads to networks that are hard to train.
> > >
> > > I still stand by my remark that the main point of the lottery ticket paper is not speeding up training.

---

> > > > ### Author Response · Authors · 2020-11-25
> > > > **Re: A few things to note**
> > > >
> > > > > First, [8] was not cited in [1] and [9] is a later work, so [1] did not treat [8,9] as a baseline to improve upon.
> > > >
> > > > [8] is cited in [1] in Sections 1, 2, 5, and 7, for a total of 8 citations throughout the paper. [9] is a comparison between a generalization of techniques proposed in [1] and prior techniques (e.g., those of [8]) showing that the ultimate accuracy/sparsity tradeoffs achieved by [1] are also achievable with the theoretically slower classical pruning techniques like [8].
> > > >
> > > > > The point isn't about speeding up training, it is that the fast method leads to networks that are hard to train.
> > > >
> > > > This is essentially the same point that we make when we cite [1] as an example of a paper that "propose[s] techniques to speed up the training of efficient neural networks." By showing that it would be possible for the fast method to train accurate networks, rather than the inaccurate networks observed in prior work, [1] in effect (and in stated motivation) proposes speeding up training of efficient neural networks.
> > > >
> > > > As you point out, there are certainly other framings of [1]'s contributions, and we do not claim that our paper is analogous in every way to [1] — just that [1] is one of many papers with the explicit stated motivation of speeding up training of efficient neural networks, providing evidence from the literature against the assertion that "...the time to train a network is of much less importance..." (R2).
> > > >
> > > > ### Citations
> > > >
> > > > [1] "The Lottery Ticket Hypothesis," Jonathan Frankle and Michael Carbin.
> > > >
> > > > [8] "Learning both weights and connections for efficient neural networks," Song Han, Jeff Pool, John Tran, William J. Dally.
> > > >
> > > > [9] "Comparing Rewinding and Fine-tuning in Neural Network Pruning," Alex Renda, Jonathan Frankle, Michael Carbin.

---

> ### Comment · AnonReviewer1 · 2020-11-20
> **Regarding Hyperparameter choices and sweeps**
>
> Thanks for the author response. Yes, I think the results for multiple datasets in Table 1 are great, and understand that it takes significant time/compute investment to generate these results.
>
> My main point, which I believe Reviewer4 shares, is that all results are focused on the PACT method. It is natural to wonder whether the results will generalize to other kinds of binarization strategies. I think it is not necessary to rerun all experiments using a different binarization method, but it would be helpful to at least show some preliminary experiments just to give more confidence to the generalizability of the results.
>
> The other choice would be to state it more explicitly in writing that these results are for PACT and discuss alternative methods in the future work/conclusion section.

---

> > ### Author Response · Authors · 2020-11-25
> > **Re: Hyperparameter choices and sweeps**
> >
> > > It is natural to wonder whether the results will generalize to other kinds of binarization strategies...
> >
> > This is a reasonable question, and indeed one that our experiments do not answer. To avoid generating potentially misleading conclusions about the performance of the method on binarization schemes not fully evaluated in the work, we will add statements to the introduction that our results are only validated with static weight scaling and PACT activation scaling.

---

### Decision · Program_Chairs · 2021-01-07
**Final Decision**

**Decision:**

Reject

**Comment:**

## Description

The paper asks the question whether it is possible to accelerate training a binarized neural network from scratch to a given target accuracy [by starting with training a full-precision network]. The main claimed contributions are: the idea to use *partially* pretrained networks, experimental evidence regarding the split of the training budget and measuring the speed-up.

## Review Process and Decision

All four reviewers agreed in the low rating of the paper and in the opinion that the paper is not a significant contribution. The area chair supports rejection.

## Details

It has been already observed that pre-training  in some form is needed for achieving the best accuracy:
Rastegari (2016) XNOR-Net: ImageNet Classification Using Binary Convolutional Neural Networks
Bulat (2019), "Improved training of binary networks for human pose estimation and image recognition"
Martinez (2020): "Training Binary Neural Networks with Real-to-Binary Convolutions"

Alizadeh, (2019 fig. 4) notice that pre-training can be viewed as a speed-up, but in their setup find that training from scratch gives a better accuracy. Bulat (2019) and Martinez (2020), on the contrary do use pre-training to achieve the best accuracy.  It is questionable whether the pre-training in these works is partial or not. I believe the result largely depends on the pre-training method used, which is not discussed in depth in the submission. More generally, some graduated optimization methods such as graduated smoothing or graduated non-convexity are known to help in finding better solutions / lead to faster optimization and in fact Bulat (2019) use pre-training with gradual transition from smooth activation to the sign function.  Relative to these points the technical contribution (one paragraph in the paper) is not significant. The empirical part of the contribution shows some effect, but does not indicate a breakthrough on its own. An investigation / design of pre-training schemes could make it more substantial.

The empirical analysis proposed does not rule out, and in fact supports, the methodology that for the best final accuracy, the full rather than partial pre-training is useful.

The gain of speed-up by a factor 1.3 (diminishing to close to 1 if we are interested in the best accuracy), is of little practical interest. In particular, a slight code optimization can give a similar speed-up without the complexity and hyperparameters involved in pre-training. The authors write  "we are not aware of any effort to exploit binarization during the training phase"
There are available public implementations that can optimize the forward pass of binary networks, in particular on GPU, while backward pass can stay in full precision. It could give a similar speed-up. In particular Courbariaux (2016) provides a GPU kernel and proposes a variant of BatchNorm with bit shifts rather than multiplications, specifically used at training time.
Making the emphasis on a relatively small speed-up that can be obtained to train sub-optimal models, in my view is not a good strategy to present this work. Rather the phenomenon that (partial) pre-training helps with the goal to improve the training methods more substantially I find of higher interest.

Finally, I agree with the reviewers that the lottery ticket hypothesis (Frankle, 2020) work speaks of the speed-up only hypothetically and its main (and fairly in-depth) contribution is in demonstrating and investigating an interesting phenomenon about training and initialization, which I do not see relevant to this submission.